# Characterization of Stressing Conditions in a High Energy Ball Mill by Discrete Element Simulations

**Christine Friederike Burmeister** [1,*] **, Moritz Hofer** [1] **, Palanivel Molaiyan** [2] **, Peter Michalowski** [1] **and Arno Kwade** [1]

1 Institute for Particle Technology, Technische Universität Braunschweig, Volkmaroder Straße 5, 38104 Braunschweig, Germany; m.hofer@tu-braunschweig.de (M.H.); p.michalowski@tu-braunschweig.de (P.M.); a.kwade@tu-braunschweig.de (A.K.)
2 Research Unit of Sustainable Chemistry, University of Oulu, P.O. Box 4300, 90570 Oulu, Finland; palanivel.molaiyan@oulu.fi
* Correspondence: c.burmeister@tu-braunschweig.de

**Abstract:** The synthesis of sulfide solid electrolytes in ball mills by mechanochemical routes not only is efficient but also can enable the upscaling of material synthesis as required for the commercialization of solid-state battery materials. On a laboratory scale, the Emax high energy ball mill accounts for high stresses and power densities, as well as for temperature control, to prevent damage to the material and equipment even for long process times. To overcome the merely phenomenological treatment, we characterized the milling process in an Emax by DEM simulations, using the sulfide solid electrolyte LPS as a model material for the calibration of input parameters to the DEM, and compared it to a planetary ball mill for a selected parameter set. We derived mechanistic model equations for the stressing conditions depending on the operation parameters of rotational speed, media size and filling ratio. The stressing conditions are of importance as they determine the outcome of the mechanochemical milling process, thus forming the basis for evaluating and interpreting experiments and for establishing scaling rules for the process transfer to larger mills.

**Keywords:** high energy ball mill; discrete element method; stressing model; sulfide solid electrolyte





## 1. Introduction

Today's use of ball mills is very promising for the synthesis of solid-state battery materials, as mechanochemical milling processes offer the potential for the synthesis of large amounts of solid electrolytes, which are required to drive forward the commercialization of solid-state batteries [1]. In contrast, the employment of the classic solid-state synthesis that is usually performed inside quartz ampoules in small quantities is challenging in terms of scalability. An upscaling of liquid-based processes may be possible [2–4], but it is accompanied by possible contamination of the solvent residues and a potentially higher impact on the environment [2].

The alternative route through mechanical mixing and milling techniques allows not only the mechanochemical synthesis of solid electrolytes [1,5] but also the processing of cathode composites, even on a large scale [6], by ensuring high intensities, which result in the required high ionic conductivities. So far, mechanochemical processes are most often limited to the laboratory scale, e.g., in planetary ball or high energy mills, and viewed solely experimentally.

When it comes to materials or processes that are sensitive to temperature effects, sufficient temperature control can be crucial to account for temperature-dependent kinetics or to prevent degradation and damage of the processed material or equipment. The Emax high energy ball mill enables the cooling of milling chambers while providing stressing energies and power densities similar to or even higher than those of planetary ball mills.

An initial description of the power input of the Emax high energy ball mill was given based on a kinematic model that was first introduced by Burgio et al. [7] and applied to the Emax by Kessler et al. [8]. The product yield of a mechanocatalytic depolymerization process was successfully correlated to the dissipated energy, while the influence of local energy dissipation and the frequency of media collisions was not further investigated.

However, the operation parameters of the mechanochemical milling process have a differentiated impact on the structure and the transport properties of ionic conductors and, thus, their resulting performance as solid electrolytes. Therefore, beyond the power input, the effect of local energy dissipation and frequency of stressing should be taken into account. In this study, we investigate these stressing conditions in an Emax high energy ball mill using discrete element method (DEM) simulations, as these conditions are not experimentally accessible. The DEM includes friction through rolling and sliding and accounts for changes in the motion pattern due to frictional effects, depending on the powder properties [9,10]. According to considerations presented in other publications [10–12], we describe these stressing conditions and their dependencies on the operation parameters of media size, $d_{meida}$; media filling ratio, $\varphi_{meida}$; and rotational speed, n and derive mechanistic model equations.

The stressing conditions within the Emax high energy ball mill are not experimentally accessible, nor have they been described systematically elsewhere in the literature. They can hardly be estimated from the process parameters. Therefore, to access the important characteristics of stressing intensity and collision frequency within this new mill type, numerical simulations using DEM are required. By considering not only the energy overall but also how often and at what intensity the material is stressed, the individual effects on the structural formation and yield can be identified. The process transfer to larger mill types, e.g., eccentric vibration mills or stirred media mills, shall be done with regard to the optimum stressing conditions, which can be configured selectively, based on either mechanistic models or DEM simulations.

The stressing conditions within a ball mill affect and determine the grinding, as well as the mechanochemical processes and the quality of the products, as demonstrated by Kwade et al., where the product fineness and yield were defined in detail [13]. Besides the type of stressing, whether the particles are arranged as single particles, in layers, or in particle beds, the product quality is affected by the stressing energy and stressing intensity, as well as the collision frequency [11–15].

The stressing energy, SE, as a mill-related characteristic parameter, is transferred to the particles during one stressing event. The ratio of stressing energy to the volume or mass of the stressed particles is defined as the stressing intensity, SI, and is a product-related characteristic parameter that describes the specific energy input at one stressing event. The second mill-related characteristic parameter is the frequency of media collisions, CF. Hence, it is immensely important to understand how the milling parameters and the stressing conditions affect the energy transfer to the solids, and the resulting reactions or micro processes in general [11–19]. This study focuses on the mill-related conditions of stressing energy and collision frequency, which are calculated using numerical simulations.

As a model material, a sulfide solid electrolyte was used, as this class of material can be synthesized via mechanochemical routes in ball mills [1,20–23]. The processing time for solid electrolytes in planetary ball mills is often excessive due to the fact that long pause times for cooling down have to be taken into account. A reduction in process times is therefore desired, and it can be achieved by means of coolable, small laboratory or larger stirred media mills.

## 2. Materials and Methods

### 2.1. Simulation Set-Up

The investigated high energy ball mill Emax (Retsch GmbH, Haan, Germany) features two chambers that move on a circular path with a radius R = 17 mm in the same rotation direction so that they are always located on the opposite side of their circular path to main-

tain the balance of forces (Figure 1b). In the simulation, the movement of a single 125 mL chamber is reproduced by superimposing two linear, orthogonal oscillations (Figure 1a), both with an amplitude of the radius R. Assuming the starting point of the movement, M, in the middle of the circle, oscillation $\theta_1$ starts first, while the second oscillation, $\theta_2$, starts with a time offset, representing the time of a quarter rotation (Equation (1)).

$$t_{offset} = \frac{0.25}{n \cdot 60} \tag{1}$$

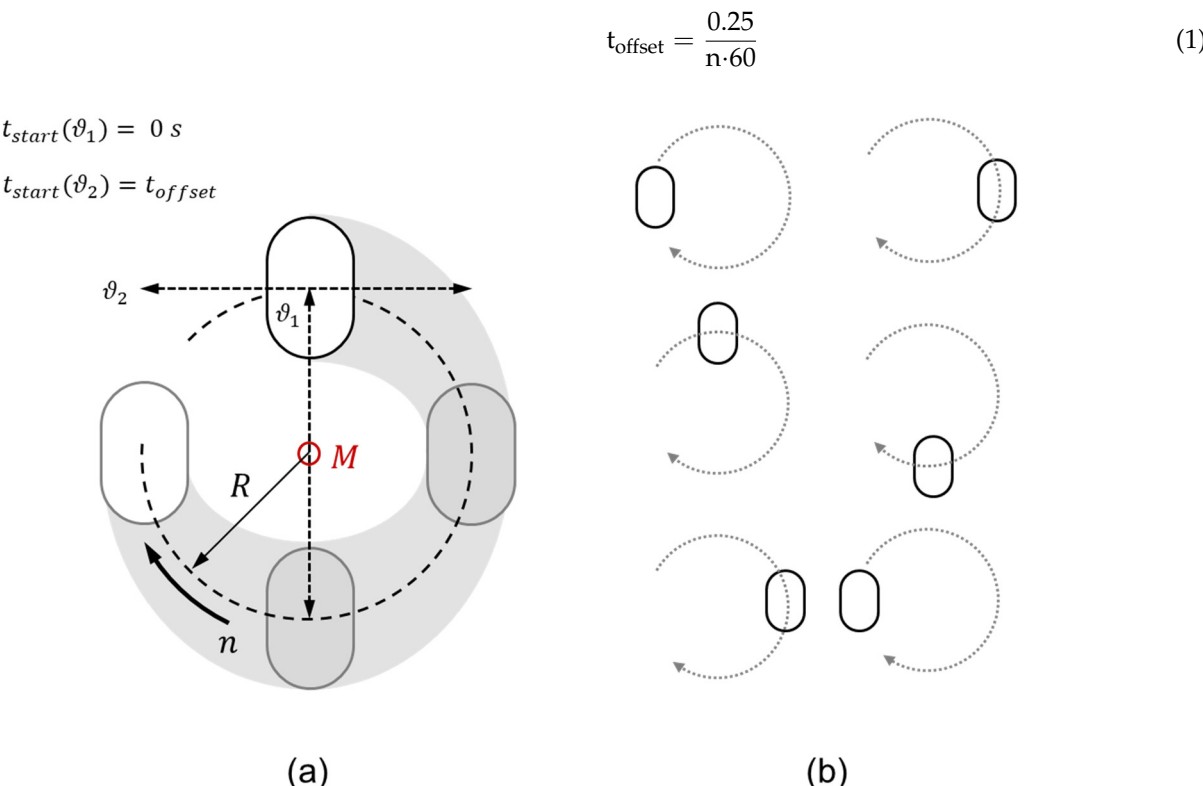

**Figure 1.** Circular movement of one milling chamber: (**a**) implementation in the DEM simulation by superimposition of linear oscillations, $\theta_1$ and $\theta_2$, R = 17 mm; (**b**) chamber positions.

### 2.2. Simulation Model

The movement of the chamber and media, and thereby the stressing conditions, are modelled based on the discrete element method [10,24], using the software package EDEM 2.7 (DEM Solutions). For each media contact, the DEM calculates the forces of compression and shear based on the contact model of Hertz [25] and Mindlin [26] and calculates the resulting velocities and accelerations according to Newton's law of motion.

In order to minimize the computational complexity, our DEM model limits the number of discrete elements for the grinding media and excludes discretization of the processed material by considering its effect by applying appropriate parameter values for friction and restitution as already shown in the literature [10,27,28]. Using a robust calibration of the friction and restitution coefficients described in the following, the simulation depicts the number of stressing events, as well as the energy dissipation on local and global levels, while the breakage and conversion of the educt material are not directly depicted. The consideration of educt material as individual particles is thus not required.

However, due to the presence of material, both as a free-flowing powder and a layer on surfaces, in the physical system, the friction and restitution behavior changes depending on the product material. Studies show that the adjustment of the coefficients of friction and restitution reproduces the system behavior and the media motion and, in doing so, enables appropriate values for stressing conditions to be obtained [10,12,27]. The authors already described the series of experiments and simulations required to adjust the coefficients of the virtual system in detail [27]: Media covered with product particles are used to execute free-fall tests, to determine the restitution. The fall and the bounce height of the media are

recorded with a high-speed camera, and the coefficient of restitution, COR, of the collision is described by the square root of the ratio of bounce height to fall height. The friction coefficients are adjusted by correlating results gained from experiments, and simulations that reproduce these experiments. First, the rolling behavior of a single grinding medium inside a grinding chamber, both coated with material, is examined. The rolling behavior is simulated in DEM while the coefficient of rolling friction, $\mu_R$ is varied until both position curves coincide. The material coverage on the medium and the walls retards the rolling motion, so the number of peaks is characteristic for a certain friction value [27,28]. The rolling behavior remains almost unaffected by the static friction, $\mu_S$ that is examined in the next step by the angle of repose. The angle is measured experimentally and afterwards modeled in DEM for varied coefficients of static friction. Both values are finally compared, and the correct friction coefficient is selected for matching values of angle of repose in simulation and experiment [12,16].

As the material covers the surfaces of the chamber wall and the grinding media, it affects the media–wall and the media–media contacts in the same manner. The contact and material parameters for the DEM model are shown in Tables 1 and 2.

**Table 1.** Contact parameters for the DEM model.

| Model Parameter | Calibrated Input Value |
|---|---|
| Coefficient of Rolling Friction $\mu_R$ | 0.05 |
| Coefficient of Static Friction $\mu_S$ | 0.58 |
| Coefficient of Restitution COR | 0.47 |

**Table 2.** Material parameters for the DEM model for the grinding media and milling chamber, both made from zirconia.

| Parameter | Input Value |
|---|---|
| Shear modulus G | $8 \cdot 10^{10}$ Pa |
| Poisson's ratio $\nu$ | 0.3 |
| Density $\rho$ | 5900 kgm$^{-3}$ |

As a tryout material to determine the input parameters of the DEM simulation, we used the sulfide solid electrolyte $Li_6PS_5Br$ (density 1.9 g/cm$^3$). The material was prepared by mechanochemical milling in a high-energy planetary ball mill (Pulverisette 7, Co. Fritsch GmbH, Idar-Oberstein, Germany) with zirconia grinding media in a 125 mL chamber made also made from zirconia. The starting materials were mixed with a stoichiometric amount of lithium disulfide ($Li_2S$, Sigma-Aldrich Chemie GmbH, Taufkirchen, Germany), phosphorus(V) sulfide ($P_2S_5$, Sigma-Aldrich Chemie GmbH, Taufkirchen, Germany) and lithium bromide (LiBr, Thermo Fisher (Kandel) GmbH, Kandel Germany) at a rotation speed of 510 min$^{-1}$ for 10 h, which was 30 cycles of 10 min rest and 10 min milling time. All educts were used as received.

As the time step, the critical Rayleigh time step [29] was chosen, which is calculated from the minimal particle radius, the minimal density of the used materials, the maximum shear modulus $G_{max}$ and Poisson's ratio $\nu_{max}$. The minimal particle size equals the media radius $r_{media}$, as only one media size is used for each simulation.

$$t_{crit} = \frac{\pi r_{media} \cdot \sqrt{\frac{\rho_{min}}{G_{max}}}}{0.1631 \nu_{max} + 0.8766} \qquad (2)$$

The contact parameters are average values and depend on the behavior of the contact surface covered with particles. Although the coefficients depend on the material, its hardness, plasticity and particle size, for a contact system of media covered with particles, similar values can be found for other systems, such as for the comminution

of marble [10] and alumina [12] and the mechanochemical synthesis of 5-(4-hydroxy-3-methoxybenzylidene)barbituric acid [16]. Additionally, the system is not extremely sensitive to the model parameters, especially when the operation parameters are varied, and so it is assumed that all materials of comparable particle size and properties, e.g., other materials within the thiophosphate solid electrolyte material class, are represented by these values.

### 2.3. Simulation Validation via Thermal Energy

The experimental milling set-up of the Emax high energy mill is not equipped with a torque sensor, to enable the direct measurement of the power input. An electrical measurement may give an estimation of the power draw but is subject to the influences in the control of the complex drive mechanics, and thus a direct correlation between electric energy consumption and energy dissipation in the grinding chamber cannot be ensured.

Instead, the heat dissipation to the internal cooling system is considered, to derive the power draw experimentally, which is a less accurate but simpler method. As cooling aid, 3 L of water was used (mass of fluid $m_f$ = 3 kg, $c_P$ = 4182 J kg$^{-1}$ K$^{-1}$), which was pumped through the internal cooling system of the mill in a circuit at a pump frequency of 300 min$^{-1}$. The temperature was measured on-line in the pump receiver tank (Huber Pilot ONE) for a processing time of 30 min.

### 2.4. Simulation Operation Parameters

The operation parameters under investigation are the size of media, $d_{media}$, made out of zirconia; the media filling ratio, $\varphi_{media}$; and the rotational speed, n. The rotational speed was varied between 600 and 1200 min$^{-1}$. The chamber had a volume of 125 mL and was also made from zirconia.

The media filling ratio (Equation (3)) is defined as the ratio of media bulk volume, $V_{media, bulk}$, to the chamber volume, $V_{chamber}$. This definition is reasonable, as a filling higher than 1 is not possible, although free pore volume in the media bulk is still available to the processed material [1].

$$\varphi_{media} = \frac{V_{media, bulk}}{V_{chamber}} = \frac{m_{media}}{(1 - \epsilon) \cdot \rho_{media} \cdot V_{chamber}} \tag{3}$$

The bulk volume of media can be calculated based on the mass, $m_{media}$, and the density, $\rho_{media}$, of media, as well as the bulk porosity, $\epsilon$. Here, the bulk porosity of zirconia media is assumed to be $\epsilon = 0.41$ for all media sizes, which leads to the same mass of media charge, regardless of media size, if the filling is kept constant

The media filling ratios and the corresponding media numbers are presented in Table 3.

**Table 3.** Emax—media filling for $V_{chamber}$ = 125 mL.

| Media Filling Ratio $\varphi_{media}$ | Media Number $N_{media}$ | | |
|:---:|:---:|:---:|:---:|
| | 5 mm | 7 mm | 10 mm |
| 0.1 | - | - | 14 |
| 0.2 | - | - | 28 |
| 0.3 | 332 | 123 | 42 |
| 0.4 | 451 | 164 | 56 |
| 0.5 | - | - | 70 |
| 0.6 | - | - | 84 |
| 0.7 | - | - | 98 |

### 2.5. Comparison of High Energy Mill and Planetary Mill

While the simulation part of this study does not include any experimental outcome of a mechanochemical synthesis, for some selected parameter settings the mechanochemical

synthesis of $Li_6PS_5Br$ was conducted in the Emax, as well as in a planetary ball mill (Pulverisette 7, Fritsch GmbH, Idar-Oberstein, Germany) with a similar size range, in order to compare these mills. The received product was characterized with regard to the ionic conductivity.

For the experiments, the media filling ratio was kept constant at $\varphi_{media} = 0.3$, and two media sizes, 5 mm and 10 mm, were tested. For comparison, an identical rotational speed of 800 $min^{-1}$ was chosen, as well as a higher rotational speed of 1200 $min^{-1}$ for the Emax, in order to adjust comparable stressing conditions in both mills. The parameter settings used for experiments and simulation are shown in Table 4.

**Table 4.** Operation parameters for the comparison of Emax and planetary mill.

| Mill | Chamber Volume/mL | Rotational Speed $n$/$min^{-1}$ | Media Size $d_{media}$/mm | Media Number $N_{media}$/– | Mass of Educts/g |
|---|---|---|---|---|---|
| Emax | 125 | 800 | 5 | 332 | 8 |
| Emax | 125 | 800 | 10 | 42 | 8 |
| Emax | 125 | 1200 | 5 | 332 | 8 |
| Emax | 125 | 1200 | 10 | 42 | 8 |
| Planetary mill | 80 | 800 | 5 | 216 | 5 |
| Planetary mill | 80 | 800 | 10 | 27 | 5 |

The starting materials were used in a constant ball-to-powder ratio and were mixed with a stoichiometric amount of lithium disulfide ($Li_2S$, Sigma-Aldrich Chemie GmbH, Taufkirchen, Germany), phosphorus(V) sulfide ($P_2S_5$, Sigma Aldrich) and lithium bromide (LiBr, Thermo Fisher (Kandel) GmbH, Kandel Germany).

The temperature of the Emax high energy mill was controlled by an internal water cooling system at a constant fluid temperature of 20 °C. The planetary ball mill does not feature active temperature control; therefore, the mill was always stopped after 5 min of processing for a 15 min break, so the overall processing time was 4 times longer. The time of mechanochemical processing was 10 h.

The ionic conductivities of the product were determined by electrochemical impedance spectroscopy (EIS), by measuring the potentiostatic impedance spectra of powder pellets (∼600 µm thickness, ∼2.01 $cm^2$ electrode area). Therefore, the synthesized powder was pelletized by uniaxial pressing at 380 MPa. The obtained pellets were placed in a Teflon tube, and stainless steel rods were used as blocking electrodes on both sides. EIS was conducted at 20 °C under 50 MPa uniaxial pressure using a Zennium potentiostat (Zahner-elektrik GmbH & Co. KG, Kronach, Germany) at frequencies from 4 MHz to 10 Hz with an amplitude of 10 mV.

### 2.6. Calculation of Energy Dissipation

Considering the stressing energy, SE, a distinction is made between normal fractions transferred by head-on collisions in the normal direction and total dissipation of the entire collision, which also includes frictional energy. With regard to particle breakage, the normal stressing energy, $SE_{normal}$, is by far the most relevant energy [12,30] and is calculated from the conservation of energy and momentum during the collision [31]. It is a function of the relative collision velocity, $v_{rel,n}$, in the normal direction, which is extracted as a simulation result, and the masses, $m_1$ and $m_2$, of the colliding media [10–12,24,31,32], as well as the coefficient of restitution. The coefficient of restitution is a dimensionless number reflecting the plastic–elastic behavior of the collision, which is thus used as a simulation input parameter, as well as for the calculation of stressing energy.

$$SE_{normal} = \frac{m_1 m_2}{2(m_1 + m_2)} v_{rel,n}^2 \cdot \left(1 - COR^2\right) \tag{4}$$

Equation (4) demonstrates that the stressing energy can be much lower than the kinetic energy of the milling media, as only part of it is actually dissipated. This is especially the case if the relative velocity between the colliding milling media is low, or the coefficient of

restitution is close to unity due to a high collision elasticity. Additionally, it already shows that the local energy dissipation scales with the mass of milling media.

The power input, P, either total or in the normal direction, is the product of the collision frequency, CF, and mean stressing energy, $\overline{SE}$ [13]. The power describes the energy that is available per unit time for the total amount of educts, and can directly limit the process rate [12,13,16]:

$$P_{normal} = CF \cdot \overline{SE}_{normal} \tag{5}$$

$$P_{total} = CF \cdot \overline{SE}_{total} \tag{6}$$

The total power, $P_{total}$ (Equations (6) and (7)), is a function of the rotational speed, n (compare Figure 1), and the torque, M, which is extracted from the respective DEM simulation [12], although it may also be experimentally measured by a torque sensor, if available. However, the investigated Emax mill is not equipped with a torque sensor, and the power input is only accessible via DEM or kinematic models.

$$P_{total} = 2\pi \cdot n \cdot M \tag{7}$$

The total stressing energy is not accessible via energy and momentum conservation, but can be obtained as a mean value, $\overline{SE}_{total}$ (Equation (8)), using the total power input and the collision frequency, which is directly extracted from the simulation [16].

$$\overline{SE}_{total} = \frac{P_{total}}{CF}. \tag{8}$$

The power values enable the calculation of the specific energy (compare Equation (9)) [13], which describes the energy demand to synthesize or process a certain product mass, $m_P$. In order to identify the optimum stressing conditions, the specific energy demand shall be compared [12,13,16], as the process time t does not necessarily correspond to the energetic optimum.

$$E_m = \frac{P}{m_P} \cdot t \tag{9}$$

## 3. Results and Discussion

### 3.1. Simulation Validation

The simulation of a milling system, in which energy dissipation takes place via individual contacts, can be validated by the comparison of experimental and calculated power input, as shown by Burmeister et al. [12]. However, the Emax high energy mill is not equipped with a torque sensor; thus, the power draw for validation uses the heat dissipation of the mill. The temperature curves over a process time of 30 min are depicted in Figure 2 for varied rotational speeds. As expected, higher rotational speeds lead to higher temperatures, indicating also higher power inputs. Each curve strives towards a plateau, so the temperature increase is reduced over time, which is attributed to the larger temperature difference between the cooling fluid and mill system and the atmosphere. The system is not adiabatic and thus exchanges heat with its environment, and a temperature equilibrium will be reached at a certain point.

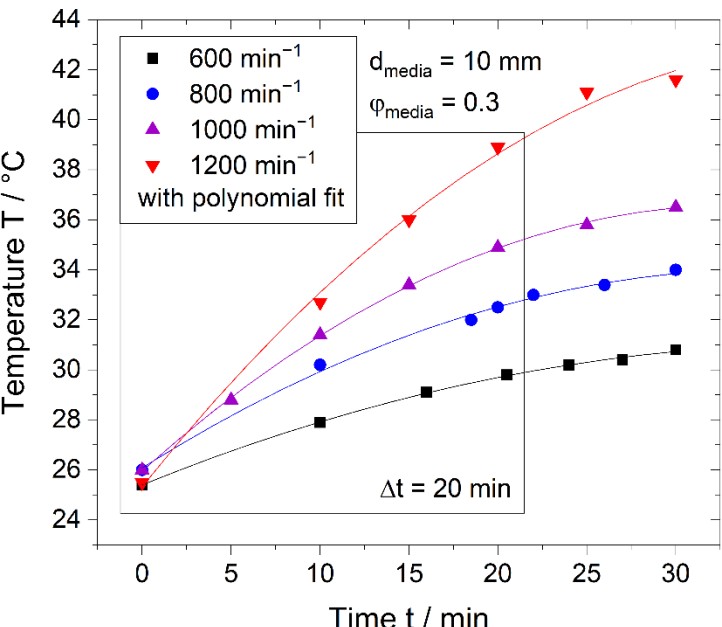

**Figure 2.** Temperature of cooling fluid for mechanochemical milling at varied rotational speeds, $\varphi_{media} = 0.3$, $d_{media} = 10$ mm.

Thus, as temperature used for the calculation of heat, Q, and thermal power, $P_Q$, (Equations (10) and (11)) only the temperature of the cooling fluid (water) within the time interval $\Delta t$ of 20 min was taken into account, to exclude the range of curve flattening.

$$Q = c_P \cdot m_f \cdot (T_0 - T_{20min}) \tag{10}$$

$$P_Q = \frac{Q}{\Delta t}, \; \Delta t = 20min \tag{11}$$

In Figure 3 the calculated thermal power from the heat experiment is compared to the power of the chamber calculated by DEM simulations. To compare the experimental and the simulation results, the experimental power from the heat measurement was divided by 2 to obtain a comparable outcome. However, the simulation still shows much lower power values, which can be attributed to the power consumption in the complex drive and the mechanics to move the vessels. This power loss within the drive and the mechanics cannot be depicted within the simulation, which only models the energy dissipation inside the chamber.

It is expected that the higher the power and thus the temperature, the more inaccurate the measurement becomes. This can explain the differences in slope between experiment and simulation, where the simulation shows a stronger increase in power with rotational speed. Although the simulation does not perfectly fit the experimental data, the comparison shows a reasonable agreement, and the heat measurement could also easily be applied to other systems when simulations are not applicable or available.

### 3.2. Effect of Media Size and Rotational Speed

Both parameters, the media size and the rotational speed, not only are easy to adjust, but also have a direct effect on the stressing conditions. However, the appropriate selection of each of these parameters may facilitate the generation, interpretation and discussion of experimental results. First, we show the effect on the power input, which is depicted in Figure 4 as a function of rotational speed, for different media sizes at a filling ratio of 0.3. The power input, both for collisions in total and only in the normal direction, is mainly determined by the rotational speed, while the media size shows a minor effect.

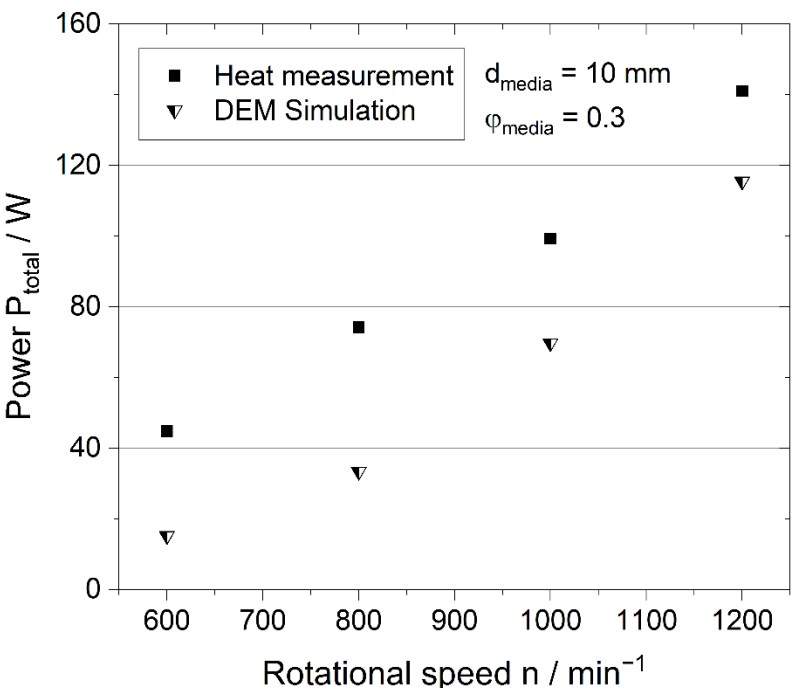

**Figure 3.** Comparison of power calculated from heat measurement and DEM simulation (one vessel). Temperature taken into account for heat calculation within the first 20 min of measurement (compare Figure 2), $\varphi_{media}$ = 0.3, $d_{media}$ = 10 mm.

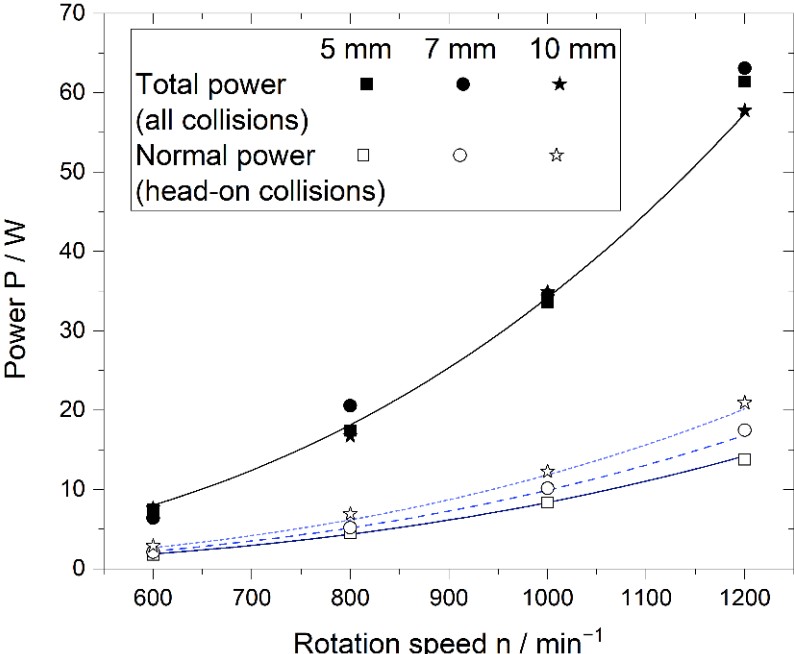

**Figure 4.** Total and normal power input in dependence on rotation speed for varied media sizes, $\varphi_{media}$ = 0.3 (fit functions are based on Equations (12) and (13)).

That means, that in terms of evaluation, one can vary the media size without changing the power input significantly, in order to investigate the effect of stressing energy and the number of stressing events.

In contrast, the variation of speed can reveal how the process, be it a comminution or a mechanochemical synthesis, is affected by the power input. For the chosen setting, with

constant friction and damping parameters, the following model equations can be derived (Tables 5 and 6):

$$P_{total} = k_{P,t} \cdot n^{a_{P,t}} \tag{12}$$

$$P_{normal} = k_{P,n} \cdot n^{a_{P,n}} \cdot d_{media}{}^{b_{P,n}} \tag{13}$$

**Table 5.** Variables for model equation (Equation (12)) to describe the total power input based on rotational speed n (in $min^{-1}$).

| Filling Ratio $\varphi_{media}/-$ | Coefficient $k_{P,t}/-$ | Exponent $a_{P,t}/-$ |
|---|---|---|
| 0.3 | $1.06 \times 10^{-7}$ | 2.84 |
| 0.4 | $1.06 \times 10^{-7}$ | 2.86 |

**Table 6.** Variables for model equation (Equation (13)) to describe the normal power input based on rotational speed n (in $min^{-1}$) and media size $d_{media}$ (in mm).

| Filling Ratio $\varphi_{media}/-$ | Coefficient $k_{P,n}/-$ | Exponent $a_{P,n}/-$ | Exponent $b_{P,n}/-$ |
|---|---|---|---|
| 0.3 | $6.75 \times 10^{-9}$ | 2.91 | 0.50 |
| 0.4 | $6.75 \times 10^{-9}$ | 2.96 | 0.44 |

The fitted data show that there is no systematic effect of the media size on the total power input (Equation (12)), and, for an increase in filling ratio, slightly higher powers are reached due to a higher exponent for the rotational speed (an increase from 2.84 to 2.86).

In contrast, the normal power input is affected by the media size and the rotational speed (Equation (13)). The exponent is in a similar range (2.91 for $\varphi_{media}$ = 0.3 and 2.96 for $\varphi_{media}$ = 0.4), while the exponent for the media size lies around 0.5, indicating that for smaller media the fraction of normal power is reduced compared to the total power.

Similar to this approach, Kessler et al. [8] developed a kinematic model to calculate the power input. The calculation uses the energy dissipation of head-on collisions and does not include friction, so only the normal power is considered, which shows a comparable dependency on the rotation speed of $n^3$, and a slight effect of media diameter. However, in contrast to our results, the kinematic model results in higher power input of 5 mm over larger media of 10 mm.

Contrary to the effect on the power, the media size strongly affects the stressing energy, as depicted in Figure 5 for the normal stressing energy at a media filling ratio of 0.3. Coarser media not only dissipate larger energies due to the higher mass (according to Equation (4)), but the dependencies (Equations (14) and (15)) also show that coarser diameters favor a regime that leads to even higher dissipation than derived from the mass increase, resulting in an exponent larger than 3. Not shown are the results for the total stressing energy at a filling of 0.3, as well as the stressing energies at a filling of 0.4, as the dependencies are almost identical: the total stressing energy is always higher than the normal, and increasing the higher filling ratio shifts the curves to lower values, which is discussed in more detailed in the corresponding section below (Tables 7 and 8).

$$\overline{SE}_{normal} = k_n \cdot n^{a_n} \cdot d_{media}{}^{b_n} \tag{14}$$

$$\overline{SE}_{total} = k_t \cdot n^{a_t} \cdot d_{media}{}^{b_t} \tag{15}$$

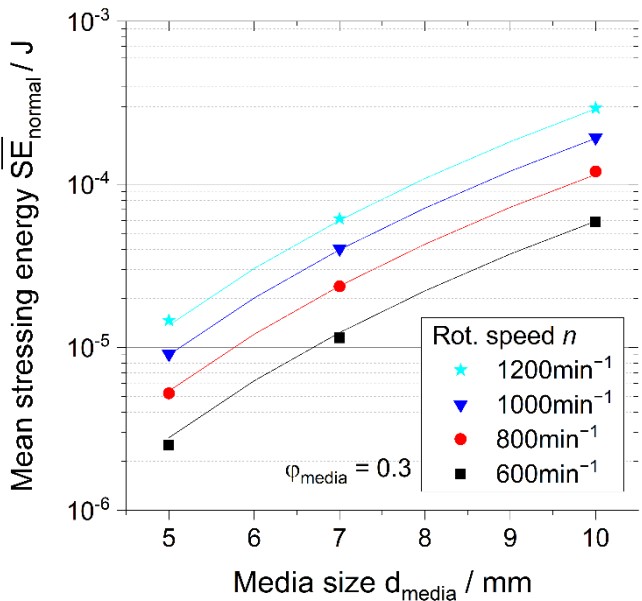

**Figure 5.** Mean stressing energy of head-on collisions as a function of media size for varied rotational speeds (fit functions are based on Equation (13)).

**Table 7.** Variables for model equation (Equation (14)) to describe the normal stressing energy based on rotational speed n (in $\text{min}^{-1}$) and media size $d_{media}$ (in mm).

| Filling Ratio $\varphi_{media}/-$ | Coefficient $k_t/-$ | Exponent of Rotational Speed $a_t/-$ | Exponent of Media Size $b_t/-$ |
|---|---|---|---|
| 0.3 | $1.00 \times 10^{-15}$ | 2.29 | 4.41 |
| 0.4 | $7.00 \times 10^{-16}$ | 2.29 | 4.39 |

**Table 8.** Variables for model equation (Equation (15)) to describe the total stressing energy based on rotational speed n (in $\text{min}^{-1}$) and media size $d_{media}$ (in mm).

| Filling Ratio $\varphi_{media}/-$ | Coefficient $k_t/-$ | Exponent of Rotational Speed $a_t/-$ | Exponent of Media Size $b_t/-$ |
|---|---|---|---|
| 0.3 | $6.90 \times 10^{-15}$ | 2.37 | 3.76 |
| 0.4 | $4.90 \times 10^{-15}$ | 2.32 | 3.91 |

The impact of media size on the normal stressing (Equation (14)) is larger than on the total stressing energy (Equation (15)), which indicates the enhanced dissipation by means of head-on collisions. The rotational speed influences the normal and the total stressing energy to almost the same extent, so it can be stated that the higher speeds not only increase the impact velocity but also affect the energy dissipation, due to sliding and friction, in the same manner.

At a constant filling ratio, the use of coarser media consequently results in a smaller number of media and, eventually, in smaller collision frequencies (Figure 6). The reduction does not linearly decrease with media number but follows an exponential function that can be expressed as a function of media number or media diameter. Due to the virtually constant power input, the collision frequency is reduced almost to the same extent as the stressing energy is increased (compare Equation (16) to Equation (15)).

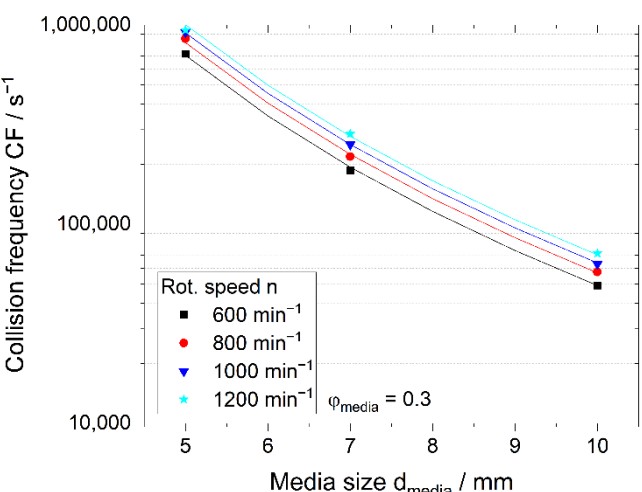

**Figure 6.** Collision frequency as a function of media size for varied rotation speeds, $\varphi_{media}$ = 0.3 (fit functions are based on Equation (16)).

Higher rotational speeds induce higher collision frequencies, which can be attributed to the shorter time interval between the collisions, due to the higher media velocities. However, compared to the effect of media size, the rotational speed has a rather minor influence (Table 9).

$$CF = k_{CF} \cdot n^{a_{CF}} \cdot d_{media}^{-b_{CF}} \tag{16}$$

**Table 9.** Variables for model equation (Equation (16)) to describe the collision frequency based on rotational speed n (in min$^{-1}$) and media size $d_{media}$ in mm).

| Filling Ratio $\varphi_{media}$/− | Coefficient $k_{CF}$/− | Exponent of Rotational Speed $a_{CF}$/− | Exponent of Media Size $b_{CF}$/− |
|---|---|---|---|
| 0.3 | $1.28 \times 10^7$ | 0.51 | 3.84 |
| 0.4 | $2.59 \times 10^7$ | 0.53 | 3.99 |

In comparison to a simulative description of a planetary ball mill given by Burmeister et al. [12] (also at a media filling ratio of 0.3, but in a 250 mL chamber), the influence of the media size is in a very similar range: $d_{media}^{4.33}$ and $d_{media}^{3.85}$ in the planetary mill, compared to $d_{media}^{4.41}$ and $d_{media}^{3.76}$ in the Emax for the normal and total stressing energy, respectively. However, the exponent, and thus the effect of the rotational speed, is much lower for the Emax, with exponents of $n^{2.29}$ and $n^{2.37}$ for normal and total stressing, while the planetary mill reaches exponents of $n^{2.68}$ and $n^{2.67}$, respectively. For the description of collision frequency, the exponents for both the Emax and the planetary mill lie in the same range.

### 3.3. Effect of Media Filling Ratio

The media filling ratio affects the stressing energy and the collision frequency. As for a specified size, a larger number of media leads to exponentially more contacts occurring, however, often with lower stressing energy.

The collision frequency in Figure 7 rises with higher fillings, though the slope of the curve decreases with the increase in the filling. Relative to the media number, it can be seen that the specific frequency (per media) is still increased, and the number effect is overproportioned, but not strongly. An exception is the very low filling ratio of 0.1, which has comparatively high absolute and specific frequencies.

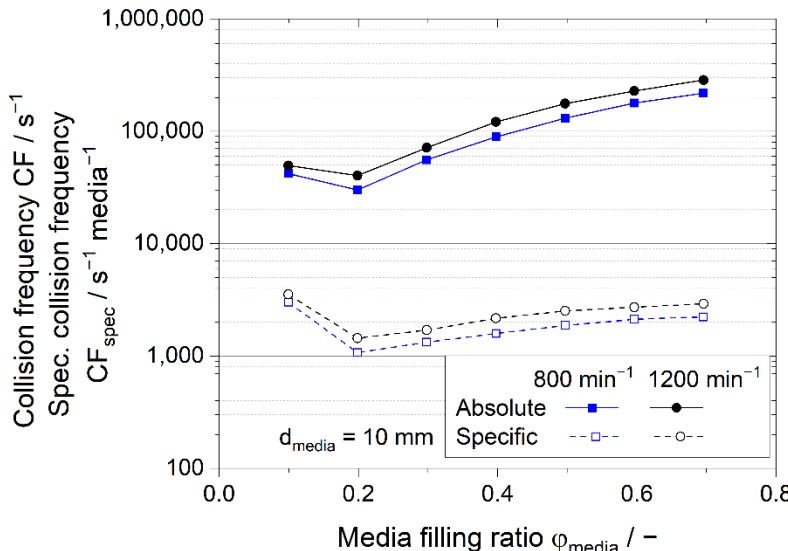

**Figure 7.** Collision frequency in dependence on media filling ratio, shown as absolute and normalized (per media) values; $d_{media}$ = 10 mm.

The stressing energy, as shown in Figure 8, is connected to the collision frequency, as it develops in the opposite direction, with decreasing stressing energy for increasing filling ratio. When the media size and rotational speed are kept constant, the energy dissipation is affected by the moving of the charge and the interaction of media. At larger collision frequencies, the free moving path of the media or the time interval between the collisions is shorter, and the energy dissipation is lower.

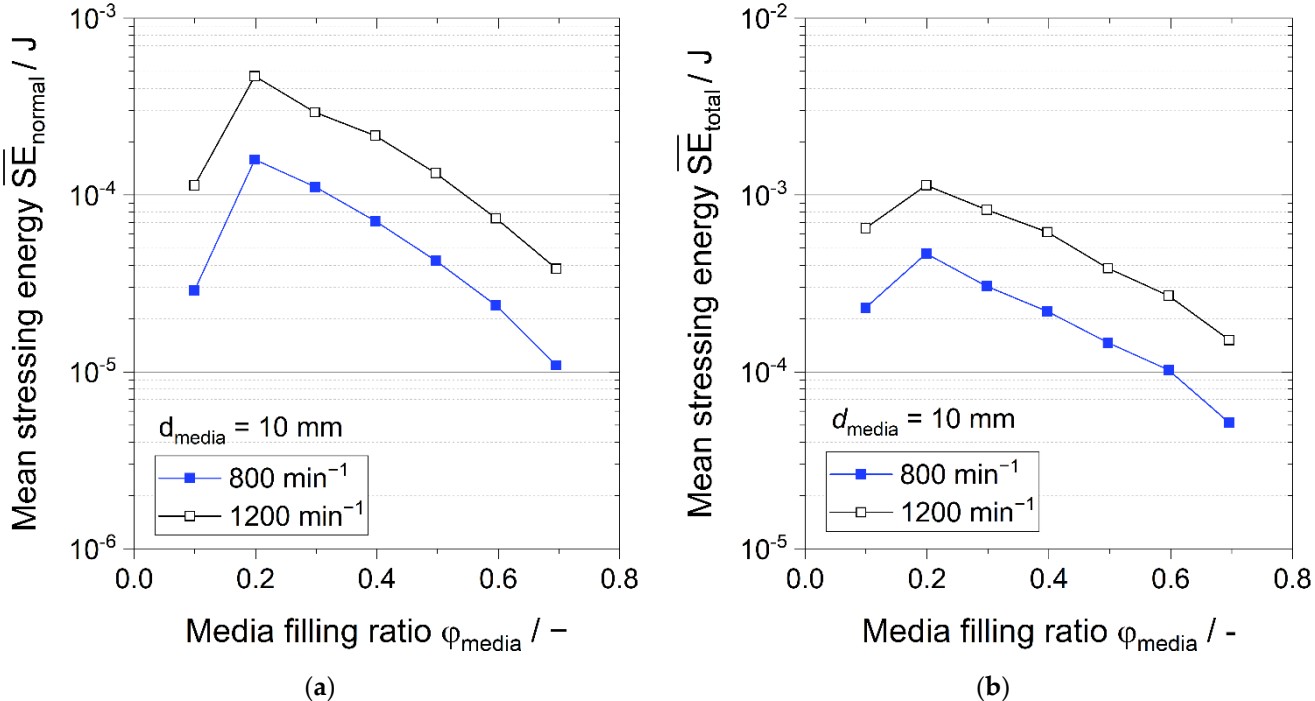

(**a**)

(**b**)

**Figure 8.** (**a**) Normal mean stressing energy and (**b**) total mean stressing energy in dependence on media filling ratio; $d_{media}$ = 10 mm.

Excluding the values at a filling of 0.1, model equations are derived to describe the decrease in normal and total stressing energy and the increase in specific collision frequency with the increase in filling ratio (Equations (17)–(19), Tables 10–12):

$$\overline{SE}_{normal} = k_{filling,n} \cdot \varphi^{f_n} \tag{17}$$

$$\overline{SE}_{total} = k_{filling,t} \cdot \varphi^{f_t} \tag{18}$$

$$CF_{spec} = k_{filling,CF} \cdot \varphi^{f_{CF}} \tag{19}$$

**Table 10.** Variables for model equation (Equation (17)) to describe the normal stressing energy based on the filling ratio $\varphi_{media}$ (−) for $d_{media}$ = 10 mm.

| Rotational Speed n /min$^{-1}$ | Coefficient $k_{filling,n}$/− | Exponent of Filling Ratio $f_n$/− |
|---|---|---|
| 800 | $1.08 \times 10^{-5}$ | 1.95 |
| 1200 | $3.16 \times 10^{-5}$ | 1.84 |

**Table 11.** Variables for model equation (Equation (18)) to describe the total stressing energy based on the filling ratio $\varphi_{media}$ (−) for $d_{media}$ = 10 mm.

| Rotational Speed n /min$^{-1}$ | Coefficient $k_{filling,t}$/− | Exponent of Filling Ratio $f_t$/− |
|---|---|---|
| 800 | $3.08 \times 10^{-4}$ | 1.00 |
| 1200 | $2.80 \times 10^{-4}$ | 0.87 |

**Table 12.** Variables for model equation (Equation (19)) to describe specific collision frequency based on media filling ratio $\varphi_{media}$ (−) for $d_{media}$ = 10 mm.

| Rotational Speed n /min$^{-1}$ | Coefficient $k_{filling,CF}$/− | Exponent of Filling Ratio $f_{CF}$/− |
|---|---|---|
| 800 | $3.22 \times 10^3$ | 0.69 |
| 1200 | $3.68 \times 10^3$ | 0.59 |

However, the collision frequency is increased to a greater extent than the stressing energy is reduced, resulting in an increase in absolute power input. Both the normal and the total power input increase with higher fillings until a maximum is reached at a filling ratio of 0.4.

The subsequent decrease is attributed to the hindered media motion: compared to lower fillings, the media cannot move freely and the media velocity is reduced. However, when conducting an experimental study in ball mills, mostly the ball-to-powder ratio or powder filling ratio is kept constant, which means that with higher media filling the amount of powder is also increased accordingly. To make a statement about the energy available to the powder, we calculated the specific power as a ratio of absolute power to the number of media. Higher specific powers indicate that, relatively speaking, more energy is available to the powder at a constant powder filling ratio.

Considering the normal energy dissipation, the highest normal stressing energy in Figure 8 is reached at a filling of 0.2, but the absolute normal power input in Figure 9 shows its maximum at a filling of 0.3–0.4. The consideration of the specific energy dissipation of head-on collisions shifts the maximum to a filling of 0.2, which corresponds with the normal stressing energy maximum.

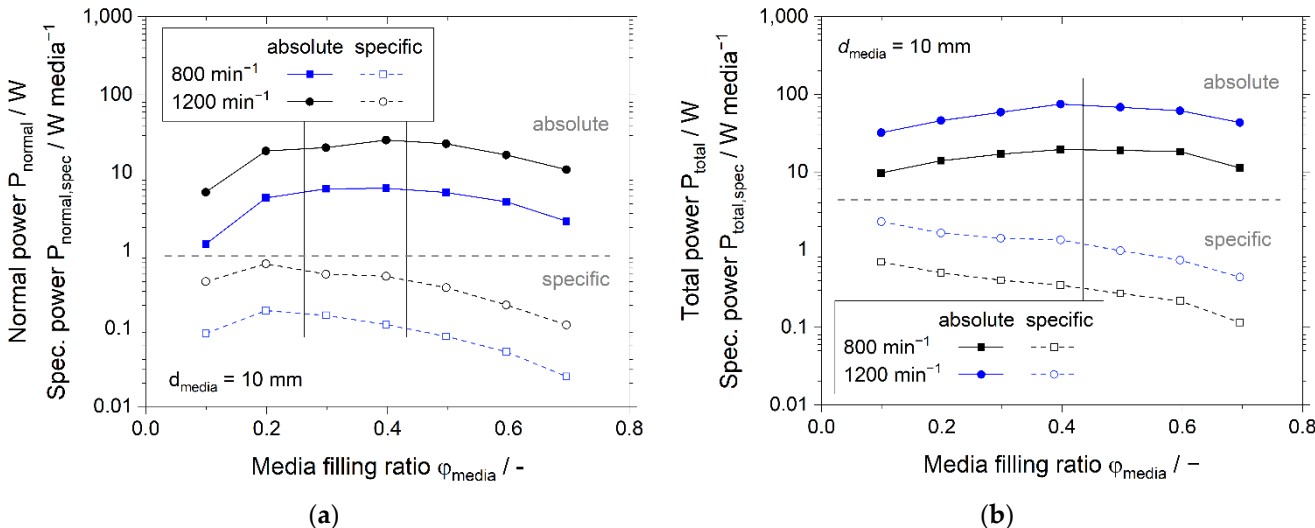

**Figure 9.** (**a**) Normal power and specific normal power (per media) in dependence on media filling ratio; (**b**) total power and specific total power (per media) in dependence on media filling ratio; $d_{media}$ = 10 mm, $P_{spec} = \frac{P}{N_{media}}$.

When the total power input is considered, the absolute maximum can be reached at fillings of 0.4, while relative to the media, the highest power corresponds to the highest specific collision frequency at the lowest media filling ratio of 0.1. Although low fillings appear beneficial due to the high stressing energies and high specific power values, the operation at the point of maximum absolute power should be considered: for the investigation of a mechanochemical process, the adjustment of stressing energies should be done via media size rather than filling ratio.

Additionally, it can be stated that the power input is subject to rather small changes when the filling ratio is changed. At low fillings, the relatively small number of media could lead to an insufficient mixing of educt material and an inhomogeneous stressing, while at the same time the amount of processed material is low, assuming a constant ball-to-powder ratio.

### 3.4. Discussion on Application of Mechanistic Model Equations

The presented model equations describe the characteristic parameters mean stressing energy and collision frequency at typical parameter settings. According to the stressing model proposed by Kwade [13,33–35], the knowledge of these two parameters enables the interpretation and prediction of favorable process conditions. Moreover, this makes the dominating stressing mechanisms accessible. Different mechanisms can play a role, depending on the process: Comminution processes with the aim of particle breakage exhibit an optimum of energy dissipation on the local level resulting in a minimum overall specific energy. If the collision frequency and, thus, the power input are maximized at this optimum stressing energy, the highest production capacity at minimum specific energy input can be achieved [12,13,34,35]. The energy needed for particle breakage is naturally dependent on the particle size and the particle strength, so the optimum is shifted to lower stressing energies for smaller sizes, but not linearly, as the particle strength increases [35].

In contrast, in some mechanochemical processes, the power input connected with the resulting increase in temperature dominates the process and the effect of stressing energy on the production rate is extremely small [16]. In this case, the stressing energies and collision frequencies resulting in the maximum power input are advantageous. However, in the mechanochemical synthesis of solid electrolytes, there are requirements for not only high yields of the conversion but also certain structural properties such as crystallinity and grain size, which are affected by the stressing energy [1,36] and influence the exhibited conduction

properties. Therefore, as the first experimental results show, power and temperature, as well as the intensity of stressing, need to be considered.

The stressing conditions can be controlled and their effect can be viewed in a suitable experimental setting: By increasing the rotational speed, stressing energy, collision frequency and power input are increased at the same time. In contrast, increasing the media size leads to higher stressing energies but lower collision frequencies and a nearly constant power input. Thus, complementary experiments enable distinguishing between the effects of each stressing condition, and the other way around; knowing the optimum stressing energy, the collision frequency for the highest productivity can be calculated [1,12,13].

While experimental studies often refer to the duration of a process, energy-dependent processes should be viewed in regard to the energy dissipation within the micro processes of breakage and chemical conversion and the overall energy demand. As shown, comparable process parameters do not result in comparable stressing conditions for different mills. Consequently, when processes are transferred from one mill to another, simulations reveal the process parameters required to keep the stressing conditions constant during the transfer.

### 3.5. Comparison to a Planetary Ball Mill

In order to evaluate the results, the mechanochemical synthesis of $Li_6PS_5Br$ was conducted for selected operation parameters in the Emax (125 mL per chamber), as well as in a planetary ball mill (80 mL per chamber).

In a first step, the same parameter setting was chosen for both mills, and the results show larger power densities, as well as higher stressing energies and collision frequencies, within the 80 mL planetary chamber (Figures 10 and 11), which consequently result in a better outcome, represented by larger conductivities (Figure 12).

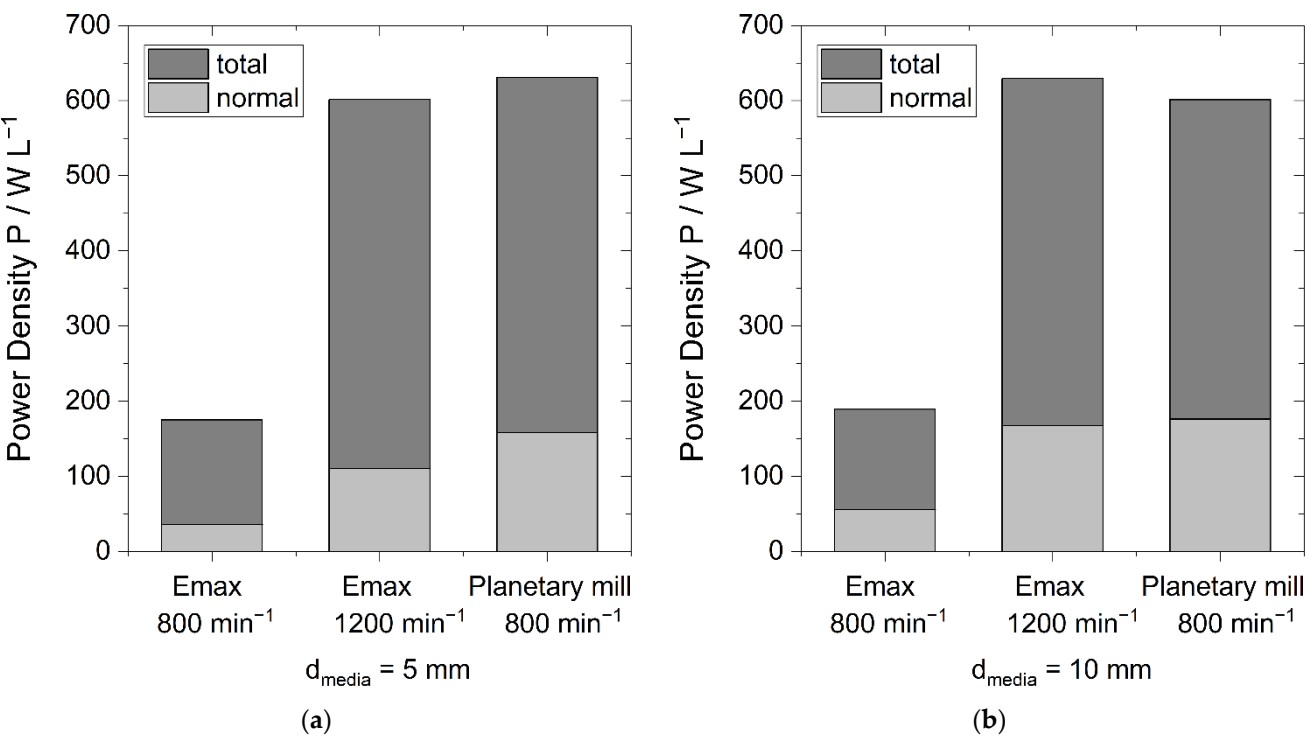

**Figure 10.** Comparison of power densities for Emax (n = 800 min$^{-1}$ and n = 1200 min$^{-1}$) and planetary ball mill (n = 800 min$^{-1}$), $\varphi_{media}$ = 0.3: (**a**) media size 5 mm; (**b**) media size 10 mm.

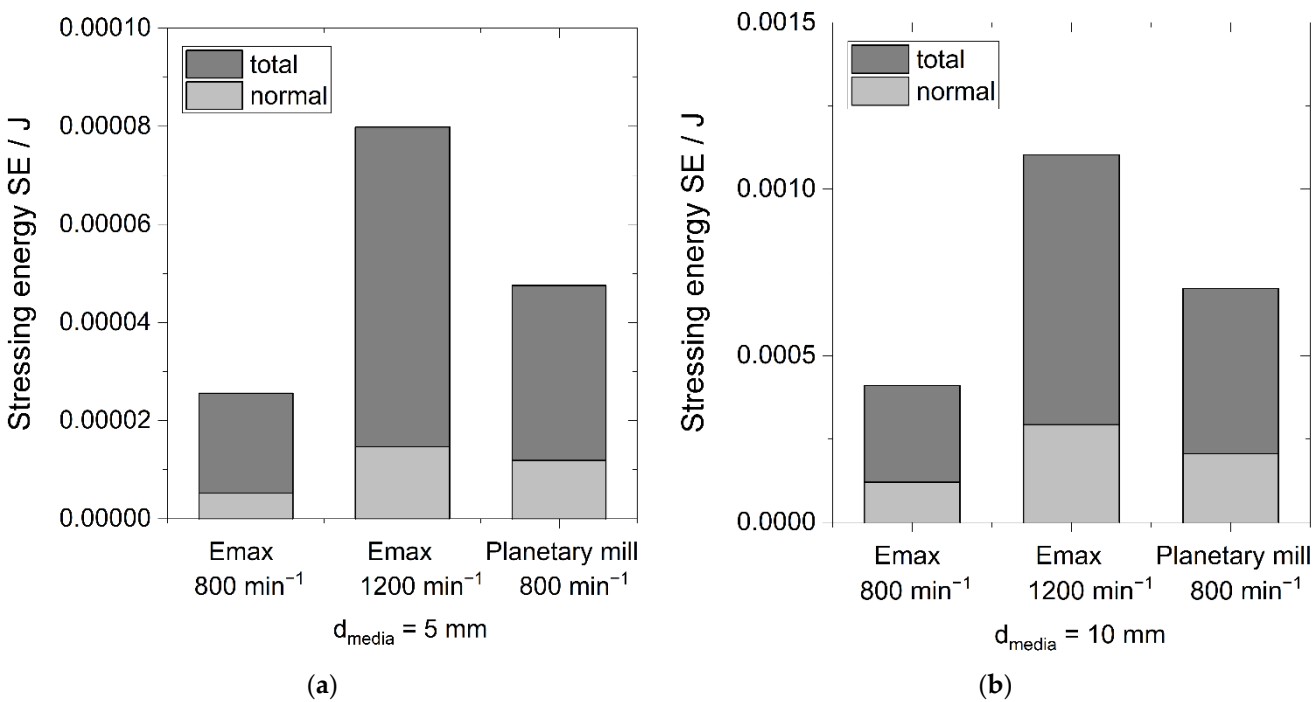

**Figure 11.** Comparison of total and normal stressing energies for Emax ($n = 800$ $min^{-1}$ and $n = 1200$ $min^{-1}$) and planetary ball mill ($n = 800$ $min^{-1}$), $\varphi_{media}$ = 0.3: (**a**) media size 5 mm; (**b**) media size 10 mm.

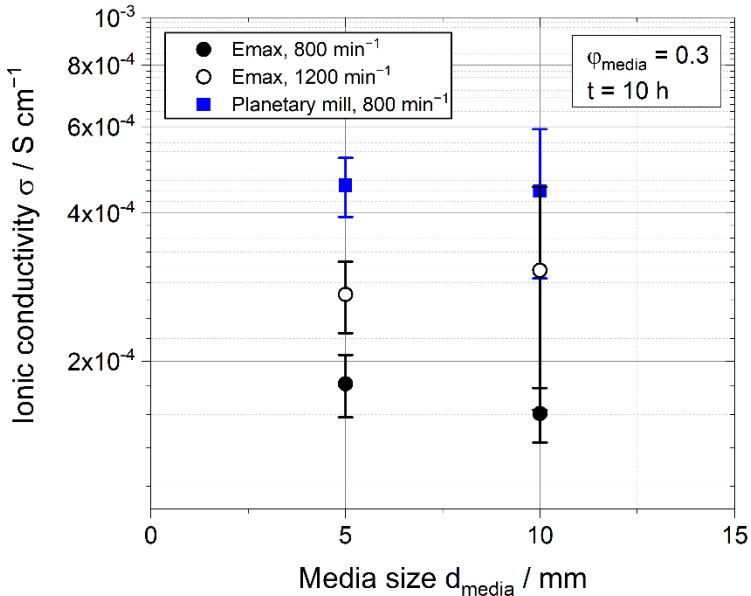

**Figure 12.** Ionic conductivity of $Li_6PS_5Br$ after 10 h of mechanochemical treatment in an Emax ($n = 800$ $min^{-1}$ and $n = 1200$ $min^{-1}$) and planetary ball mill ($n = 800$ $min^{-1}$), $\varphi_{media}$ = 0.3.

A comparable power input within the Emax and the planetary ball mill is reached when the Emax is operated at higher rotational speeds of 1200 $min^{-1}$ (Figure 10). Here, the planetary ball mill again exhibits slightly higher material conductivities; thus, the reaction is expected to be faster, and the planetary mill is beneficial to the process. The main reason should be the much higher collision number, as the stressing energy in the planetary mill is lower at a similar energy and power input. Thus, altogether almost the same amount of energy is dissipated, but the product is stressed more often. Additionally, it has to be taken into account that the planetary ball mill is not equipped with a cooling system,

and temperature control is only based on cooling breaks, resulting in a relatively high temperature. Consequently, the most probably higher temperature of the planetary ball mill should enhance the reaction and thus enhance the ionic conductivity of the resulting product, as was reported for other mechanochemical reactions [16,37,38]. A closer look at the effect of reaction conditions will be part of further studies focusing on the experimental part of the mechanochemical treatment of ball mills.

Although the planetary mill can be seen as favorable in comparison to the Emax high energy mill at the chosen parameter setting, the Emax has advantages, especially regarding temperature control. It features larger chambers, as well as an internal cooling system, so without needing extra time for cooling breaks, the absolute processing time is shortened significantly, which is particularly relevant at large processing time (here 10 h). Although larger temperatures might be advantageous with regard to the reaction, cooling is always required at long processing times to prevent the milling equipment from being damaged.

## 4. Conclusions

This study focused on a simulation-based description of the stressing conditions in the Emax high energy ball mill. Due to high stresses, in combination with the possibility of cooling, the mill is particularly suitable for mechanochemical syntheses, which offer great potential in the large-scale mechanochemical production of solid electrolytes, so the solid LPS electrolyte ($Li_6PS_5Br$) was chosen as the model material. The simulation takes the effect of the powder into account using the model parameters of friction and restitution, which were calibrated based on experiments using powder-covered grinding media. The simulation was successfully validated by the comparison of simulated power and the power calculated from the heat dissipation of the mill.

Based on the simulations, mechanistic model equations were derived to describe the relation of stressing conditions and operation parameters, such as rotational speed, media size and media filling ratio. These model equations enable the estimation of the stressing conditions for sulfide solid electrolytes, thus enabling the identification of favorable conditions.

At constant media filling, the collision frequency decreases significantly with increasing media size, due to the lower number of media. At the same time, the coarser and, thus, heavier media dissipate much higher stressing energies. This results in a power input that is only slightly affected by the media size. A large effect on the power input is gained by the rotational speed, as both stressing energy and collision frequency increase with higher rotational speeds.

The media filling ratio also affects the stressing conditions, albeit to a lesser extent: higher fillings, and thereby a higher mass of filling, can dissipate more energy. At the same time, higher fillings limit the pathways of the media, which collide in shorter intervals, resulting in increasing collision frequencies but decreasing stressing energies. Above a media filling ratio of 0.4, the effect of stressing energy diminution dominates, resulting in an absolute power maximum at a media filling of 0.4. However, the specific power values that consider the energy dissipation per media reach a maximum at low fillings due to the high stressing energies.

However, the operation at the point of maximum absolute power should be considered: for a mechanochemical process, the variation of stressing energies is recommended based on the investigation of the effect of different media sizes.

For a selected parameter setting the mechanochemical synthesis of LPS electrolyte was conducted in the Emax, as well as in a planetary ball mill with similarly sized chambers. Although operated with the same specific power input, the solid electrolyte processed in the planetary mill exhibited higher ionic conductivities, which can be attributed to higher collision frequencies, as well as higher temperatures, which might both be beneficial to the product properties. However, a more detailed view of the complex effect of the stressing conditions will be given in future studies, which will correlate the experimental results with the applied stressing conditions. The aim is to identify which conditions

are advantageous to reach high yields and at the same time favor structural properties exhibiting high ionic conductivities.

**Author Contributions:** Conceptualization, C.F.B. and M.H.; methodology, C.F.B. and M.H.; validation, C.F.B. and M.H.; formal analysis, C.F.B.; investigation, C.F.B., M.H. and P.M. (Palanivel Molaiyan); resources, M.H. and C.F.B.; data curation, C.F.B.; writing—original draft preparation, C.F.B., M.H. and P.M. (Palanivel Molaiyan); writing—review and editing, A.K.; visualization, C.F.B.; supervision, A.K.; project administration, P.M. (Peter Michalowski); funding acquisition, P.M. (Peter Michalowski) and A.K. All authors have read and agreed to the published version of the manuscript.

**Funding:** The research was supported by the Federal Ministry of Education and Research (BMBF) within the project FEST-BATT under grant number 03XP0177C.

**Institutional Review Board Statement:** Not applicable.

**Informed Consent Statement:** Not applicable.

**Data Availability Statement:** Not applicable.

**Conflicts of Interest:** The authors declare no conflict of interest.

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
