# Peer review of "Characterization of Stressing Conditions in a High Energy Ball Mill by Discrete Element Simulations"

_processes, doi:10.3390/pr10040692_

Round 1
Reviewer 1 Report
This paper can be accepted after some revisions or reply.
There are some questions or problems:
1. There are many formulas without sources.
2. The novelty of the paper could be emphasized.
3. The simulation results need to be verified by the field experiments.
4. A reference can be cited, which is on stress performance, stress test and reducing stress of a metal forming machine: Static and dynamic analysis of a 6300 KN cold orbital forging machine [J]. Processes, 2021, 9(1):7.
5. The part of conclusions is too long.
6. Simulation points in some simulations are not enough, it is not rigorous.
Reviewer 2 Report
1) The opening paragraph in the “introduction” is not making much sense. Seems like some kind of instruction. Please delete.
2) This work presents a good application of DEM where the authors use the DEM simulations to extract certain variables which cannot be measured experimentally. I think it will improve the quality of the paper if the authors also contrast their work with what already exists in the literature and point out to the uniqueness of the work. Usually with “application only” kind of paper, it is difficult to understand the novelty of the work, so I suggest that the authors explain it in their introduction.
3) Please explain what you mean by "In order to minimize the computational complexity, our DEM model limits the number of discrete elements to the grinding media, and excludes discretization of the processed material"? How do you capture the processed material in EDEM then? What does exclude discretization mean? Are you assuming the processed material as a continuum?
4) The authors present some details on the mill chamber but what about the grinding medium? Are you using the same model parameter values for particle-particle, particle-wall and wall-wall interactions? It requires more information and clarification. It is not clear to me how the authors are capturing the processed medium (see my previous comment). How about providing the geometry dimensions of the grinding medium, particle size ranges of processed medium?
5) Why are other simulation parameters not provided? Like shear modulus, Poisson’s ratio, time steps etc? How was numerical stability ensured? Were there any numerical stability issues? If yes, how were they mitigated?
6) I understand that the current work is mainly simulation based, which is fine. However, I wonder if the authors can add some discussion around how the simulation results can be used to make decisions on a real system? For example, how can we determine what’s the optimum number of grinding media for a given powder system? How will their findings change if the material properties change?
Round 2
Reviewer 1 Report
The authors made a good revision according to the revision recommendations, and I suggest that this article could be accepted directly.